# Exploring ENSO-Induced Anomalies over North America in Historical and Future Climate Simulations That Use HadGEM2-ESM Output to Drive WRF

Tristan Shepherd [1],* , Jacob J. Coburn [1], Rebecca J. Barthelmie [2] and Sara C. Pryor [1]

1   Department of Earth and Atmospheric Sciences, Cornell University, Ithaca, NY 14853, USA
2   Sibley School of Mechanical and Aerospace Engineering, Cornell University, Ithaca, NY 14853, USA
*   Correspondence: tristan.shepherd@cornell.edu

**Abstract:** Projected changes to the El Niño Southern Oscillation (ENSO) climate mode have been explored using global Earth system models (ESMs). Regional expressions of such changes have yet to be fully advanced and may require the use of regional downscaling. Here, we employ regional climate modeling (RCM) using the Weather Research and Forecasting (WRF) model at convection-permitting resolution and nested in output from the HadGEM2 ESM. We quantify ENSO teleconnections to temperature and precipitation anomalies in historical and future climate scenarios over eastern North America. Two paired simulations are run, a strong El Niño (positive ENSO phase) and a weak La Niña (negative ENSO phase), for the historical and future years. The HadGEM2 direct output and HadGEM2-WRF simulation output are compared to the anomalies derived from the NOAA ENSO Climate Normals dataset. The near-surface temperature and precipitation differences by ENSO phase, as represented by the HadGEM2-WRF historical simulations, show a poor degree of association with the NOAA ENSO Climate Normals, in part because of the large biases in the HadGEM2 model. Downscaling with the WRF model does improve the agreement with the observations, and large discrepancies remain. The model chain HadGEM2-WRF reverses the sign of the ENSO phase response over eastern North America under simulations of the future climate with high greenhouse gas forcing, but due to the poor agreement with the NOAA ENSO Climate Normals it is difficult to assign confidence to this prediction.

**Keywords:** El Niño Southern Oscillation; regional climate modelling; weather research and forecasting model; global climate modelling; climate change; convection-permitting



## 1. Introduction

The El Niño Southern Oscillation (ENSO) is a prominent mode of internal climate variability under which sea surface temperatures (SSTs) in the southern Pacific cycle between El Niño and La Niña conditions on a time scale of 2–7 years [1]. It is often referred to as a coupled climate phenomenon because changes in both SST and atmospheric circulation are required to induce a particular ENSO phase [1]. El Niño events are associated with anomalously high SSTs and convection in the eastern Pacific and descending air flow over the western Pacific. This causes a reversal of the normal circulation over the tropical Pacific [1,2]. Conversely, La Niña is associated with anomalously high SSTs and convection over the western Pacific, anomalously low SSTs in the eastern Pacific, and thus strengthening of normal atmospheric flow patterns. The transition to a specific ENSO phase most commonly emerges in the boreal autumn, with a peak in the Northern Hemisphere winter [3].

ENSO has important ramifications for temperature and precipitation anomalies across a wide range of countries, particularly during the Northern Hemisphere winter [4]. Because of the global impact of ENSO teleconnections, it is the leading source of forecast skill on seasonal to interannual time scales [5]. Past research has established relationships

between ENSO phase and temperature and precipitation patterns for countries such as the United States of America (US) [6–11]. Previous research has also sought to advance understanding of the relationships between ENSO and global climate anomalies, and how these teleconnections have changed over the last 30–40 years [9] and may evolve in the future. Future projections have used output from various generations of ESMs [12,13] and include simulations with prescribed SSTs [14]. Comparisons of temperature and precipitation composites during 1915–1960 versus 1962–2006 indicate that the north–south dipole in temperature anomalies that occurs over North America during warm ENSO winters is weakening [9]. There is emerging evidence that the ENSO-forced Pacific–North American climate mode may shift east [15] and that ENSO-derived precipitation variability may increase in many regions [16]. Recently, a new dataset (NOAA ENSO Climate Normals) of temperature and precipitation anomalies over the contiguous United States (CONUS) that accounts for background climate change trends has been developed [10].

ENSO representation in different generations of global climate and Earth system models (ESMs) has been widely evaluated. The Coupled Model Intercomparison Project 5 (CMIP5)-generation ESMs exhibited wide diversity in terms of the fidelity with which ENSO is reproduced in the historical climate [17]. The CMIP6-generation ESMs generally indicate improved fidelity in reproducing internal climate modes [18]. Other research has evaluated how ENSO teleconnections may evolve in the future [9,19–21]. An ensemble of Max–Planck ESM simulations from CMIP5 was evaluated to examine how regional manifestations of ENSO may change under a globally warmed world [22]. The projected changes in teleconnections were shown to not be statistically significant; however, in cases where the model correctly reproduced the historical climate, future temperature and precipitation anomalies were amplified under La Niña conditions [22]. Under a high greenhouse gas forcing scenario (RCP 8.5), CMIP5 ESMs manifest strong and more positive anomalies in mean daily precipitation over the southeast US, with a northward shift in the location of the maximum anomalies [23]. In CMIP6-generation ESMs, many characteristics of future ENSO remain uncertain, with some ESMs exhibiting ENSO phases that are similar to those in the observational record (CESM, FGOALS), while others exhibit phases that are too intense and persistent (ECEARTH, GISS) [18,24]. There is, however, evidence that extreme rainfall from El Niño events may double to one event per 10 years [25], which is further confirmed in CMIP6 [24,26]. It is now apparent that the models that are best able to reproduce the ENSO dynamics show an increase in future ENSO SST variability and an eastward shift and intensification of ENSO teleconnections [19]. Numerous studies have shown that ENSO projections and their global teleconnections are influenced by the degree of external (greenhouse gas) climate forcing [27–31].

Given the importance of internally forced climate variability to regional climates over North America [32] and the potential for large-scale climate change, we are motivated to address whether externally forced climate change will modify ENSO teleconnections and climate anomalies over North America. At least some fraction of the mechanisms responsible for generating temperature and/or precipitation anomalies under different ENSO phases is likely to be dictated by processes manifest at scales below those adequately treated by coarse-resolution ESMs. For example, the study domain used here covers areas of North America that exhibit very high rates of atmosphere–surface coupling [33], and precipitation regimes that are strongly influenced by mesoscale convection [34,35]. Hence, we employ an ESM-regional climate model (RCM) model chain, wherein the RCM (the Weather Research and Forecasting model, WRF) is applied at convection-permitting resolution to examine results of the differences in near-surface climate anomalies over North America between ENSO climate modes for historical and future years. We focus on maximum and minimum 2-m temperature and precipitation, but also report information regarding water vapor mixing ratio, radiation, and cloud fraction.

The Hadley Centre Global Environment Model version 2 (HadGEM2) from CMIP5 [36] is selected as the driving ESM due in part to reports that it shows a relatively good representation of the wintertime precipitation teleconnections over North America [21] and

is representative of the mean CMIP5 multi-model ensemble of ENSO conditions (i.e., frequency of positive- and negative-phase ENSO) in the historical and future climate [20]. It is also noteworthy that HadGEM2 is characterized by a relatively strong transient and equilibrium temperature response (+4.6 K) to increasing greenhouse gas concentrations ($2 \times CO_2$) [37].

The present study is the first step towards assessing: (1) whether downscaling of HadGEM2 substantially impacts ENSO phase linkages to near-surface climate anomalies over eastern North America, relative to those directly from HadGEM2, and to what degree either HadGEM2 or the HadGEM2-WRF model chain correctly characterize these teleconnections for the historical climate; and (2) whether temperature and precipitation anomalies over eastern North America as a function of ENSO phase might be modified in a warmer world. Thus, our research is formulated around three linked research questions:

1. How does the use of WRF applied at a convection-permitting resolution within the CMIP5 HadGEM2 output modify the implied teleconnections for near-surface minimum and maximum daily temperature and precipitation over the eastern USA? Does this downscaling increase the degree of association with long-term observationally derived air temperature and precipitation differences?
2. Which other modeled variables are dictating the surface responses?
3. Does this model chain suggest that these surface responses to different ENSO phases may change in the future?

## 2. Materials and Methods

### 2.1. NOAA ENSO Climate Normals

In the evaluation of the HadGEM2 and HadGEM2-WRF output, we used as the reference dataset the NOAA ENSO Climate Normals [10] computed from temperature and precipitation data in the nClimGrid dataset [38]. This dataset has a horizontal resolution of ~5 km × 5 km and describes anomalies in minimum and maximum 2-m air temperature and monthly total precipitation anomalies (expressed as percent deviation from normal) as a function of the sign and magnitude of ENSO phase.

### 2.2. Weather Research and Forecasting (WRF) Model Simulations

Simulations with WRF v3.8.1 [39] were undertaken for two years for the historical climate and two future years that represent different ENSO phases. The lateral boundary conditions (LBC) were drawn from HadGEM2-ES [36] from the 'r1i1p1' experiment and the RCP 8.5 forcing that closely approximates the current greenhouse gas trajectory [40]. The historical values of the Oceanic Niño Index (ONI) from the HadGEM2 model output [41] were used to identify ENSO phase (Figure 1) and select the simulation periods. The ONI values and the Niño 3.4 SST index (Figure 1) were used to situate these selected simulation years in the overall record from HadGEM2 and confirm that we were simulating matching phases (i.e., strong El Niño and weak La Niña) in both the historical and future context. It is noteworthy that consistent with past research [41], ONI from the HadGEM2 output identifies only three La Niña phases between 2070–2099 and a preponderance of El Niño conditions. The suppression of La Niña phases in the future is consistent with previous studies that have identified El Niño phases to increase in both frequency and intensity in future climate [24,25,42,43]. For this reason, we simulate weak La Niña and strong El Niño conditions.

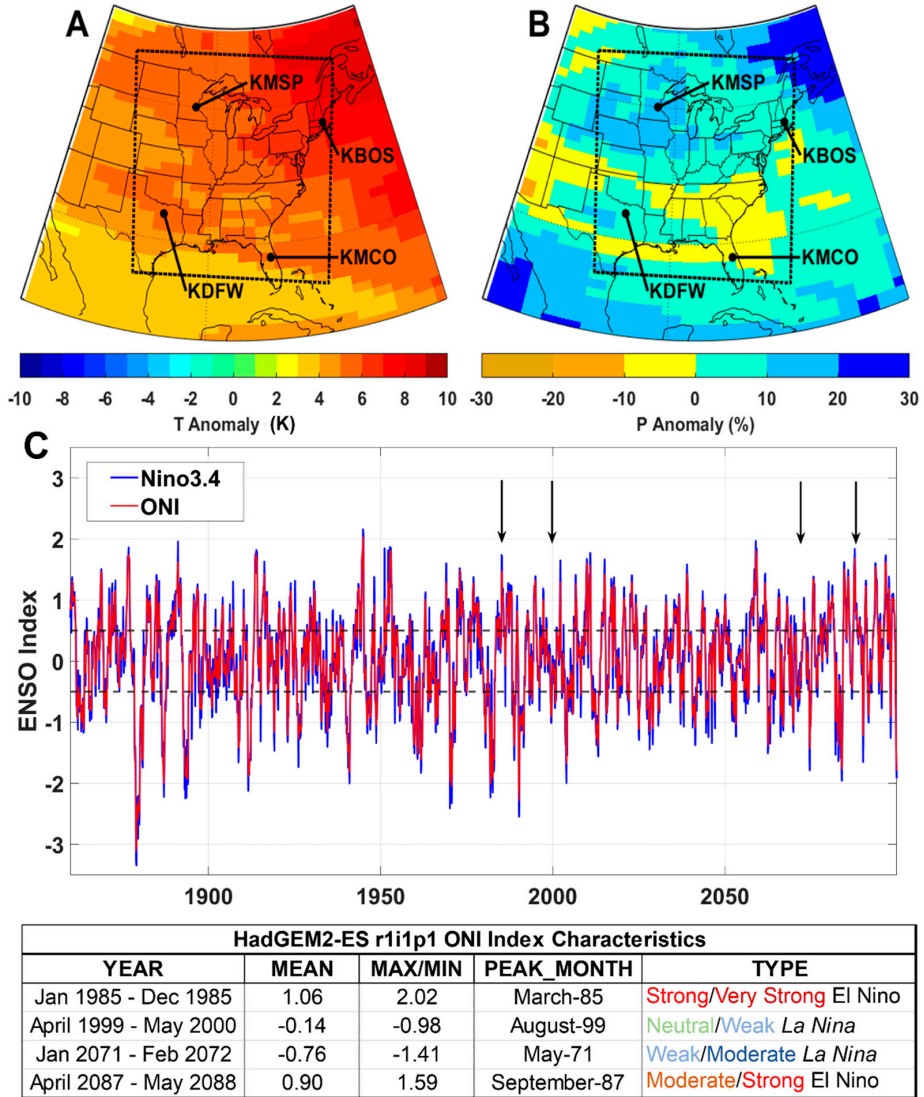

**Figure 1.** (**A**) Domains used in the WRF simulations. The outer domain comprises 320 × 320 cells at a 12 km grid spacing, while domain 2 (denoted by dashed line) comprises 676 × 676 cells at a 4 km grid spacing. The cities for which specific analyses were performed are denoted by their National Weather Service station code. Also shown is the mean temperature (T) anomaly (in Kelvin) from HadGEM2 for 2070–2099 (RCP8.5) compared to the baseline period (1970–1999). (**B**) The same as for (**A**), but with the mean precipitation (P) anomalies (as a %) from HadGEM2. (**C**) ENSO index timeseries derived from monthly SST data in HadGEM2 for the Niño3 (5–5° N, 90–150° W) and Niño3.4 (5–5° N, 120–170° W) regions. The table summarizes the Niño3 ENSO characteristics for all months within the simulation periods shown in the YEAR column, including the mean index, the relative maximum or minimum index (max for El Niño, min for La Niña), the month when the maximum or minimum occurs for each period, and the characterization of the ENSO type during each period. The arrows over the timeseries indicate the events that form the focus of this analysis (arrows from left to right match the rows in the table from top to bottom).

Representation of the impact of ENSO teleconnections on both precipitation and temperature requires both a large model domain and high resolution to include convection-permitting parameterization, all of which are very computationally demanding. Hence, two simulation periods were selected for the historical climate and the future period. The two simulated periods in the historical climate represent a strong El Niño episode (1 December 1984–1 January 1986) and a weak La Niña episode (1 April 1999–1 May 2000). The first month of each simulation is considered 'spin-up' and excluded from the analyses. WRF

simulations were also conducted for two future periods: 1 January 2071–1 February 2072 (weak La Niña) and 1 April 2087–1 May 2088 (strong El Niño). Because HadGEM2 uses a 30-day Julian calendar, each month for the WRF simulation is 30 days.

The outer model domain uses a grid resolution of 12 km with 320 × 320 cells, and the nested domain comprises 676 × 676 cells and has a grid resolution of 4 km (Figure 1). There are 41 vertical levels up to a model top of 50 hPa, and 18 of these levels are in the first 1 km of the atmosphere to suitably capture the planetary boundary layer (PBL). The HadGEM2 LBC are updated in WRF every 6 h. The key physics settings include the Eta microphysics scheme [44], rapid radiative transfer scheme for longwave radiation [45] and Dudhia for shortwave [46], Revised Monin–Obukhov similarity scheme for the surface layer physics [47], the Noah land surface model [48], the Mellor–Yamada–Nakanishi–Niino PBL scheme [49], and the Kain–Fritsch cumulus parameterization [50]. Convection is explicitly resolved in the 4 km domain, thus no cumulus scheme is applied.

Consistent with the majority of ESM-WRF simulations available in the literature ([51], including those performed within CORDEX, [52]), we use one-way coupling between the LBC from the ESM (HadGEM2) LBC and between the two WRF domains. No new ESM simulations are performed but direct output from the HadGEM2 RCP8.5 r1i1p1 experiment is also presented to contextualize that from the HadGEM2-WRF model chain. Naturally, inferences drawn from the analyses presented herein must be viewed with caution, since we consider here only two exemplar periods with different ENSO phase in the historical and future climate, and past research has shown a diversity of ENSO global teleconnections in the historical climate [10]. Nevertheless, as illustrated herein, the temperature and precipitation anomalies directly from HadGEM2 for these specific periods are similar to the mean of all events of these phases, as simulated by HadGEM2 for 1979–2009.

### 2.3. Analysis of HadGEM2-WRF Model Simulations

Unless otherwise stated, all WRF simulation output presented here are from the inner (4 km) domain. Temperature and precipitation data from the WRF model are output hourly, and the radiation variables, specific humidity, and cloud fraction data are output every 3 h. The hourly 2-m air temperatures are used to compute the daily minimum (Tmin) and maximum (Tmax) values. We focus here on two winter months, January and February (December is not evaluated because in some of the simulations this month is the following winter due to the timing of the ENSO cycle), and three summer months, June, July, and August, to provide consistency with the NOAA ENSO Climate Normals for each ENSO phase [10].

Past research has generally presented the impact of ENSO phase and intensity as air temperature and precipitation anomalies from a baseline reference period and/or the ENSO neutral phase [10]. This is not possible for the current research because we do not have a transient reference HadGEM2-WRF simulation from which ENSO neutral conditions can be derived. Thus, here we present maps (from the NOAA Climate Normals, HadGEM2, and HadGEM2-WRF) of the difference in air temperature and precipitation in the strong El Niño period minus the weak La Niña period.

The differences in Tmin, Tmax, and monthly total precipitation (PPT) deviation from the NOAA ENSO Climate Normals are computed using values in that dataset that represent a strong El Niño and a weak La Niña. The pairwise difference in mean values of daily Tmin and Tmax from HadGEM2-WRF in each grid cell and calendar month (El Niño minus La Niña) are computed and subject to a two-sample Student's *t* test to evaluate the statistical significance at the 95 % confidence level [53].

For the purposes of this analysis, WRF-simulated PPT is a monthly sum of the precipitation accumulation in each grid cell. Analyses of the PPT output at the native WRF resolution exhibit very high spatial variability in terms of the sign and magnitude of differences between ENSO phase, consistent with information that the skillful scale for precipitation even from convection-permitting-models is substantially larger than the grid spacing [54]. For this reason, and to aid with visualization and comparisons with the

direct HadGEM2 output in the maps presented here, the PPT from WRF is re-gridded from the native 4 km grid to the HadGEM2 resolution of 1.25° latitude by 1.875° longitude (equivalent to about 120 km × 139 km at 55° N). Because we do not have a sufficient baseline to compute PPT mean conditions, we express all PPT differences between the two ENSO phases in mm per month, while those in the NOAA ENSO Climate Normals are percentage deviations from the long-term mean.

Model output of four key drivers of anomalies in air temperature and precipitation are also analyzed. In each case the difference in monthly mean values is calculated as strong El Niño minus weak/moderate La Niña. The variables considered are the mean total columnar specific humidity (columnar Q); daily maximum downwelling shortwave (DSW) and outgoing longwave radiation (OLR) at the ground; and daily mean cloud fraction in low (sum of layer cloud for heights, z < 2000 m), mid (z: 2000–6000 m), and high (z > 6000 m) clouds.

## 3. Results and Discussion

### 3.1. ENSO Normals Anomalies Difference (El Niño–La Niña) in the Historical Climate HadGEM2 and HadGEM2-WRF versus the NOAA ENSO Climate Normals

The differences in the near-surface temperature (Tmin and Tmax) and precipitation shown herein are computed as the monthly mean values of each variable under a strong El Niño minus those under a weak La Niña. Examples of the monthly mean values of Tmax in January for the two ENSO phases as output from the HadGEM2 and HadGEM2-WRF models are given in Figure 2. To compute the difference fields for January in the historical climate, the spatial field shown in panel B is subtracted from that in A to give the difference based on direct HadGEM2 output, while for the HadGEM2-WRF model chain that difference is computed by subtracting the spatial field in panel D from that in panel B. As shown, while the patterns from the HadGEM2-WRF model chain are similar to those directly output from HadGEM2, there is also evidence of substantial local modification of the near-surface conditions in the downscaled output over regions with strong atmosphere–surface coupling (e.g., over the Great Lakes and in the Southern Great Plains) and of enhanced spatial variability due to improved representation of the terrain variability (e.g., in the Appalachian Mountains).

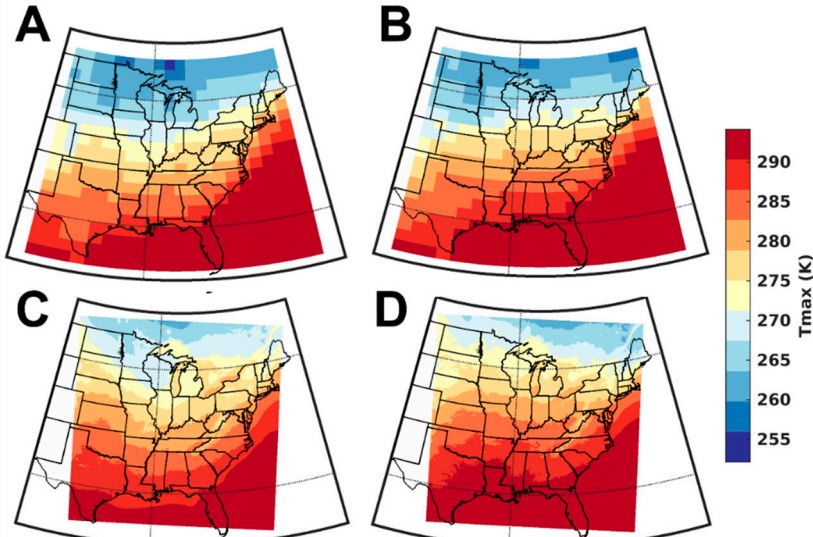

**Figure 2.** Monthly mean maximum 2-m temperature (Tmax) values for the two ENSO phases: (**A**) HadGEM2 output for January 1985 (El Niño); (**B**) HadGEM2 output for January 2000 (La Niña); (**C**) HadGEM2-WRF output for January 1985 (El Niño); (**D**) HadGEM2-WRF output for January 2000 (La Niña).

The differences in Tmin, Tmax, and monthly total precipitation under strong El Niño minus weak La Niña conditions from the NOAA ENSO Climate Normals dataset (Figure 3a, Figure 4a, Figure 5a) for winter months indicate a distinct north–south gradient in temperature differences [10]. Mean Tmax anomalies under strong El Niño conditions for January range from +1 K across the Midwest and northern states, to −1 K in the southern tier. In February, the north–south gradient shifts north with negative mean maximum 2-m temperature anomalies of −1 K prevailing over much of the Midwest [10]. For monthly mean Tmin, the negative anomalies intensify over the southern tier from January to February (−0.4 K in January to −1 K in February) [10]. The north–south gradient is biased towards positive differences in January, but in February the gradient shifts northward as the negative differences intensify. Accordingly, the difference in Tmax and Tmin strong El Niño minus weak La Niña during these winter months also exhibits a clear north–south differentiation (Figures 3a and 4a).

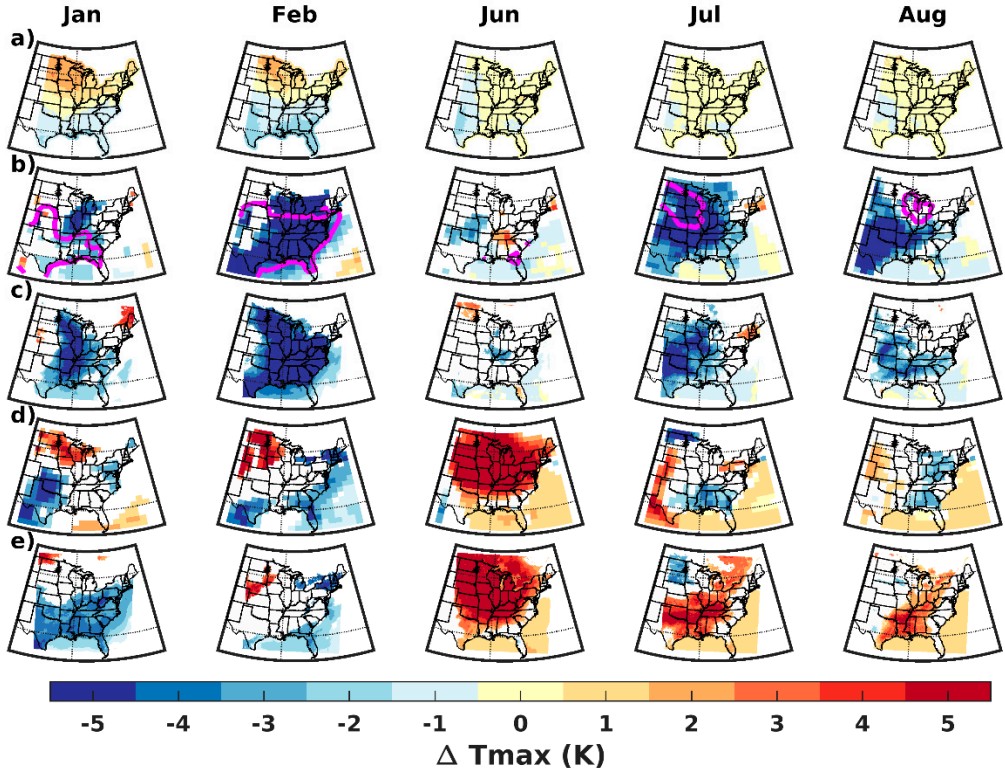

**Figure 3.** Monthly mean maximum 2-m temperature anomalies (Tmax, in Kelvin) differenced by ENSO phase (strong El Niño-weak La Niña) for the (**a**) NOAA ENSO Climate Normals dataset [10], (**b**) HadGEM2 output for the two historical years simulated using HadGEM2-WRF, (**c**) HadGEM2-WRF historical climate simulations at 4-km domain resolution, (**d**) HadGEM2 output for the two matching WRF future simulated years, and (**e**) HadGEM2-WRF future climate simulations. Differences in (**b**–**e**) are only shown for grid cells for which a two-sample *t* test indicates they are significant at a 95 % confidence interval. The magenta lines in (**b**) show the area covered by a mean Tmax difference of −0.5 °C for all strong El Niño-weak La Niña events in the historical climate, as simulated by HadGEM2.

For summer months, the gradient for the difference in Tmax and Tmin is predominantly west–east (Figures 3a and 4a). These differences are less pronounced than the anomaly maps shown in [10], because of the differencing of a strong El Niño from a weak La Niña phase. Here, the mean maximum difference is >−1 K across the southern Great Plains, with more positive (~+0.6 K) values across the northeast. For Tmin, the gradient is less pronounced, and the differences are smaller in the west (−0.4 to −0.6 K) and decreases further by August (Figure 4a).

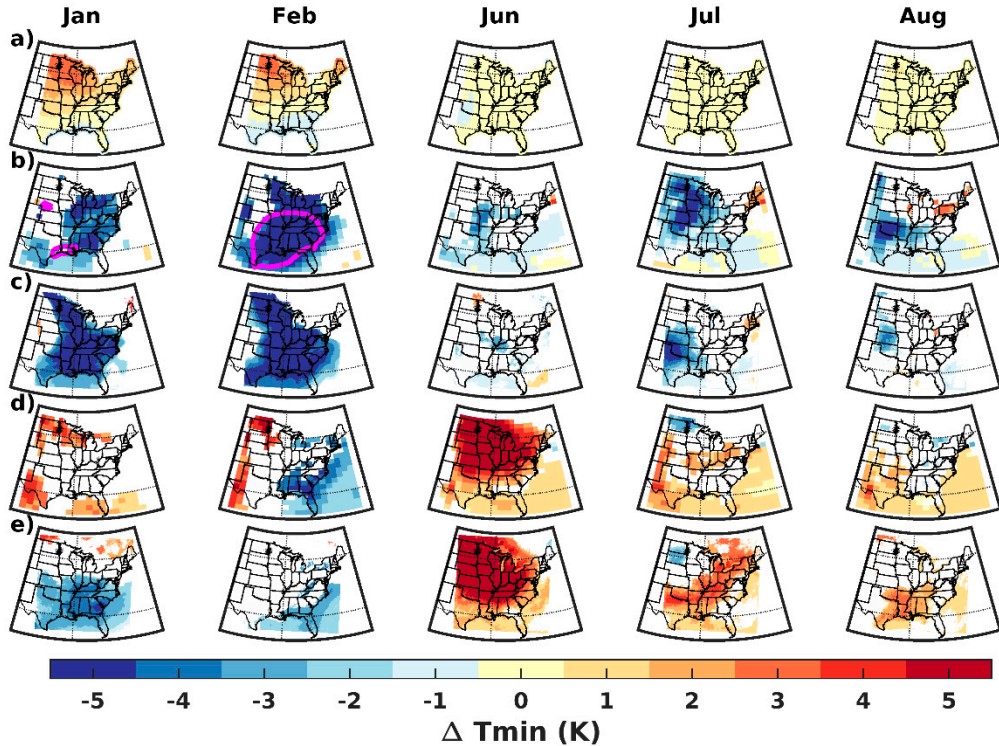

**Figure 4.** Monthly mean minimum 2-m temperature anomalies (Tmin, in Kelvin) differenced by ENSO phase (strong El Niño–weak La Niña) for the (**a**) NOAA ENSO Climate Normals dataset [10], (**b**) HadGEM2 output for the two historical years simulated using HadGEM2-WRF, (**c**) HadGEM2-WRF historical climate simulations at 4-km domain resolution, (**d**) HadGEM2 output for the two matching WRF future simulated years, and (**e**) HadGEM2-WRF future climate simulations. Differences in (**b**–**e**) are only shown for grid cells for which a two-sample *t* test indicates they are significant at a 95% confidence interval. The magenta lines in (**b**) show the area covered by a mean Tmin difference of −0.5 °C for all strong El Niño-weak La Niña events in the historical climate, as simulated by HadGEM2.

The difference in monthly accumulated precipitation in the NOAA ENSO Climate Normals in the two ENSO phases indicates relative drying and up to 50% lower wintertime PPT in the MidWest during strong El Niño versus weak La Niña, with relative wetting of the southern Great Plains and Southeast (Figure 5a). PPT differences in the summer months are less pronounced and exhibit more month-to-month variability (Figure 5a). For example, there is evidence of negative differences in monthly PPT in the early summer over the central eastern coastal states (e.g., Delaware) but positive differences in August (Figure 5a).

Comparison of the temperature difference for these specific ENSO events in the historical climate indicates that the results are similar to the average from the ensemble of strong El Niño and weak La Niña events (Figures 3b and 4b). The summertime differences in Tmin and Tmax, however, are of larger magnitude than the ensemble mean difference for strong El Niño minus weak La Niña from HadGEM2 for 1970–1999.

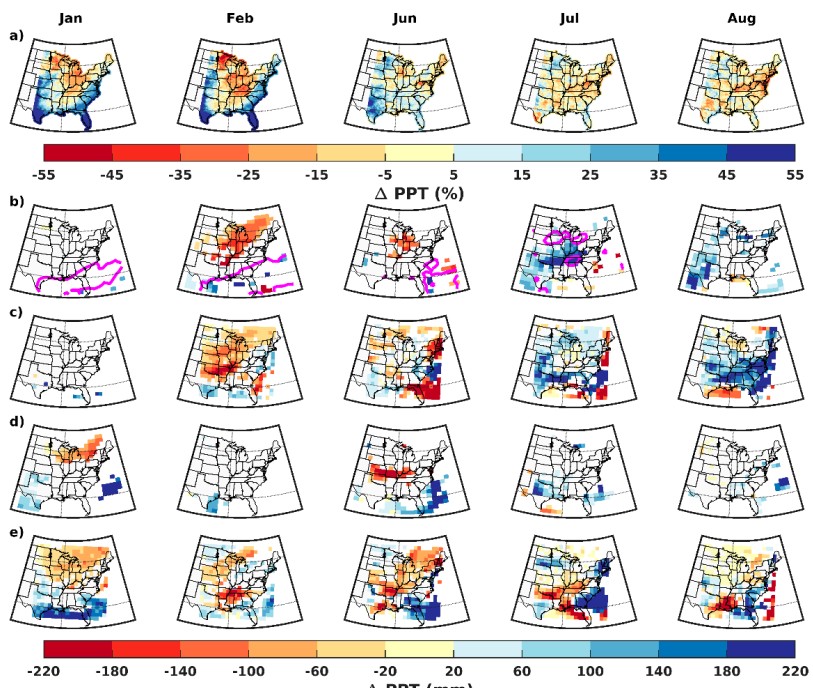

**Figure 5.** Monthly mean precipitation differences (in mm) differenced by ENSO phase (strong El Niño-weak La Niña) for the (**a**) NOAA ENSO Climate Normals dataset [10] (NB: in % deviation from the long-term mean), (**b**) HadGEM2 output for e.g. January 1985 minus January 2000 (colors: the magenta contours show the areas with +10 mm differences for all events (strong El Niño-weak La Niña) in the entire historical period (1979–2009)), (**c**) HadGEM2-WRF historical climate simulation output re-gridded to the HadGEM2 model resolution, (**d**) HadGEM2 model output for the two matching WRF future simulated years, and (**e**) the HadGEM-WRF future climate simulations re-gridded to the HadGEM2 model resolution.

Differences in mean monthly Tmin and Tmax (strong El Niño minus weak La Niña) from the NOAA ENSO Climate Normals (Figures 3a and 4a) and those from both HadGEM2 (Figures 3b and 4b) and the HadGEM2-WRF model chain (Figures 3c and 4c) in the historical climate show some similarities. For example, air temperature differences (both Tmin and Tmax) are generally of larger magnitude in the winter months. However, from Figures 3 and 4, it is evident that despite the HadGEM2 SST indices identifying the correct ENSO phase, the expected ENSO teleconnections to North America are generally not well represented. The HadGEM2 model output for the historical climate indicates that the temperature differences in the two ENSO phases generally do not agree with those from the NOAA ENSO Climate Normals dataset. The HadGEM2 output generates a much higher frequency of negative temperature differences (Figures 3b and 4b), although the summertime differences are weaker in all months. It is also evident that the failure of HadGEM2 to correctly characterize the ENSO signal cannot be corrected by downscaling using WRF. For both Tmin and Tmax differences in the historical climate (El Niño minus La Niña), direct output from HadGEM2 (Figures 3b and 4b) and the HadGEM2-WRF model chain (Figures 3c and 4c) fail to capture the expected north–south dipole in temperature differences. There is a predominance of negative difference in both mean Tmax and Tmin, and the magnitude is much larger ($<-5$ K in some grid cells) than in the NOAA ENSO Climate Normals dataset ($\sim -1$ K). The application of WRF modifies the precise spatial patterns of Tmin and Tmax differences in the winter months relative to those from HadGEM2 (Figure 2), but does not result in much closer agreement with the NOAA ENSO Climate Normals (Figures 3 and 4). In the summer months, comparison with the NOAA ENSO Climate Normals dataset (Figures 3a and 4a) shows that HadGEM2 downscaled with WRF captures some of the teleconnection signal. There are negative differences in Tmin and

Tmax values in the western portions of the southern Great Plains. The simulated positive Tmin differences over the Northeast are present in July only (Figures 3c and 4c) and are not as spatially coherent but are of a similar magnitude (+1−+2 K). In some instances where the anomalies in HadGEM2 are of a small magnitude, HadGEM2-WRF does generate a spatially coherent response of opposite sign compared to the HadGEM2 output (e.g., positive Tmax differences over Florida for the month of June (Figure 3c)).

The precipitation anomalies from the HadGEM2-WRF simulations for the historical climate correctly indicate relative drying over the Midwest during winter (Figure 5a–c) under strong El Niño conditions. For the years simulated in the historical climate, the difference in monthly total PPT values during February is lower by up to 100 mm in the strong El Niño versus weak La Niña year across much of the Midwest and Northeast in the HadGEM2-WRF output and is lower by 25% (strong El Niño versus weak La Niña) in the NOAA ENSO Climate Normals. The model chain HadGEM2-WRF manifests larger areas of positive PPT anomalies during summer than are manifest in the NOAA ENSO Climate Normals dataset and exhibits stronger deviations from the HadGEM2 direct output than is evident in the temperature anomalies. The inference is that WRF applied at convection-permitting grid spacing is notably changing the implied local PPT response to a given ENSO phase.

The historical climate HadGEM2-WRF simulations can be quantitatively compared with the NOAA ENSO Climate Normals data in terms of the joint occurrence in space of positive or negative temperature and precipitation differences (i.e., joint +/− in Figure 6). For Tmax, positive differences (strong El Niño versus weak La Niña) cover 25% and 15% of the simulated continental United States (CONUS) land area in the two winter months according to the NOAA ENSO Climate Normals (Figure 6a) and 61% and 44% of the domain for Tmin (Figure 6b). The HadGEM2-WRF output is dominated by negative Tmin and Tmax differences and thus, the common areas of positive and negative temperature differences are only 20% and 45% for Tmax and 12% and 26% for Tmax, respectively (Figure 6a,b). Areas with positive and negative PPT differences from the HadGEM2-WRF output exhibit greater agreement with those from the NOAA ENSO Climate Normals in both winter and summer months (Figure 6c).

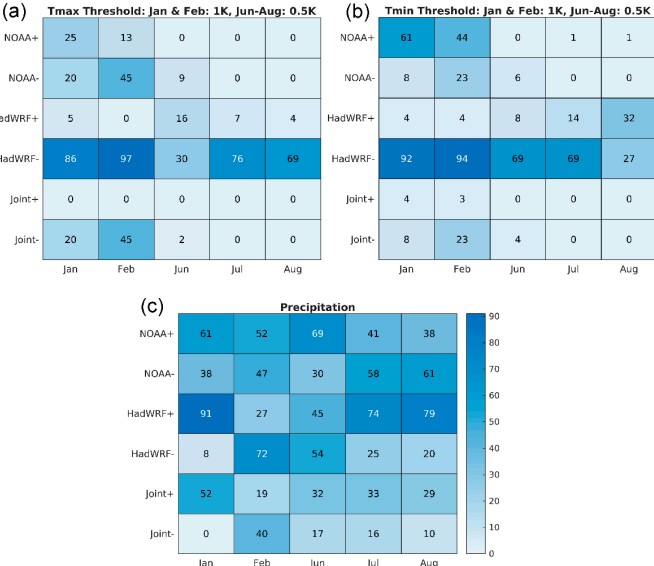

**Figure 6.** Heatmaps of the percent coverage of the WRF model domain where data from the NOAA ENSO Climate Normals are available, with positive (+) or negative (−) mean monthly (**a**) Tmax, (**b**) Tmin, or (**c**) precipitation differences; strong El Niño minus weak La Niña in the NOAA ENSO Climate Normals, the HadGEM2-WRF model chain, and in both (joint) in the historical climate. Note that in these analyses all differences from HadGEM2-WRF are evaluated at the WRF grid resolution and that thresholds used for the Tmax and Tmin differences are different in summer (0.5 K) and winter (1 K).



*3.2. Drivers of Tmax and Tmin Differences by ENSO Phase in HadGEM2-WRF*

Spearman correlation coefficients (ρ) for the spatial fields of differences in Tmin and Tmax by ENSO phase, against differences in the mean daily maximum OLR, DSW, columnar specific humidity (columnar Q), and the mean coverage of clouds at the three levels, were computed. The results show large-magnitude negative correlations between columnar specific humidity and low cloud cover and Tmax values in summer. The Spearman correlation coefficient values (ρ) = −0.64, −0.20, and −0.72 between the spatial patterns of mean Tmax differences and columnar Q differences for the three summer months. Equivalent correlation coefficients are −0.51, −0.11, and −0.47 for the spatial patterns of mean Tmax differences and differences in low cloud cover for the three summer months. This is consistent with the suggestion that areas with lower cloud cover in the strong El Niño summer months will exhibit positive temperature anomalies. The columnar specific humidity differences exhibit a positive ρ with differences in Tmax and Tmin in the winter. Spatial correlation coefficients with the radiation variables are of a smaller magnitude ($|\rho| < 0.5$ in all months).

*3.3. ENSO Impacts on Air Temperature and Precipitation: Historical and Future Climate Scenarios from HadGEM2-WRF Simulations*

Although the Tmin, Tmax, and PPT differences under strong a El Niño versus weak La Niña—as simulated by HadGEM2 and HadGEM2-WRF—show disappointingly low agreement with the NOAA ENSO Climate Normals for the historical climate, we consider how teleconnections from ENSO to temperature and precipitation over eastern North America may be modified in the future in this model chain.

The area with negative Tmax differences (strong El Niño minus weak La Niña) during the winter is much smaller in the future climate simulations based on output from either HadGEM2 or HadGEM2-WRF. Indeed, there is much greater consistency with the NOAA ENSO Climate Normals in the historical climate (Figure 3a vs. Figure 3d,e). The same is true for Tmin (Figure 4a vs. Figure 4d,e), both in terms of the sign and magnitude of difference. Recall, these are differences, so the implication is that the La Niña phases leads to warmer winter conditions in the historical climate, but cooler conditions in the future simulations (or alternatively El Niño events lead to warmer conditions in the future climate simulations). Conversely, in summer the differences in Tmax and Tmin are of opposite sign to those from the historical climate simulations (Figures 3 and 4). The strong El Niño minus weak La Niña output for the future climate simulations yields large areas with positive differences in Tmax (of >2 K) (Figure 3d,e) in June, while positive Tmax differences dominate in all three summer months, particularly in the HadGEM2-WRF output (Figure 3e). For both Tmax and Tmin, it is evident in this model chain that wintertime negative anomalies are less persistent across the continental US in the future climate simulations (*Future−* in Figure 7a,b). This is consistent with the reduced *Joint−* area of the domain coverage. The positive (*Joint+*) anomalies are in even poorer agreement—particularly for Tmax, given the strong warming signal in the future climate simulations—identified by significant domain-wide coverage in summertime months (Figure 7a). Indeed, WRF reverses the sign of the temperature anomaly in summer (from negative to positive) in the future climate simulations.

In contrast to the historical climate, the differences in monthly total PPT under the two ENSO phases in the future climate simulations show a greater dominance of relative drying under strong El Niño conditions in the HadGEM2-WRF output (Figures 5 and 7). At the regional level during the winter, drier conditions are evident in the interior and wetter conditions are evident to the south. In the summertime months, when compared to the historical climate simulations, there is enhanced precipitation across Florida in June–July, and a decrease in precipitation across areas of the southern Great Plains. This suggests that the convection-permitting WRF simulations generate a hydroclimate that is substantially different from the signal in the HadGEM2 model.

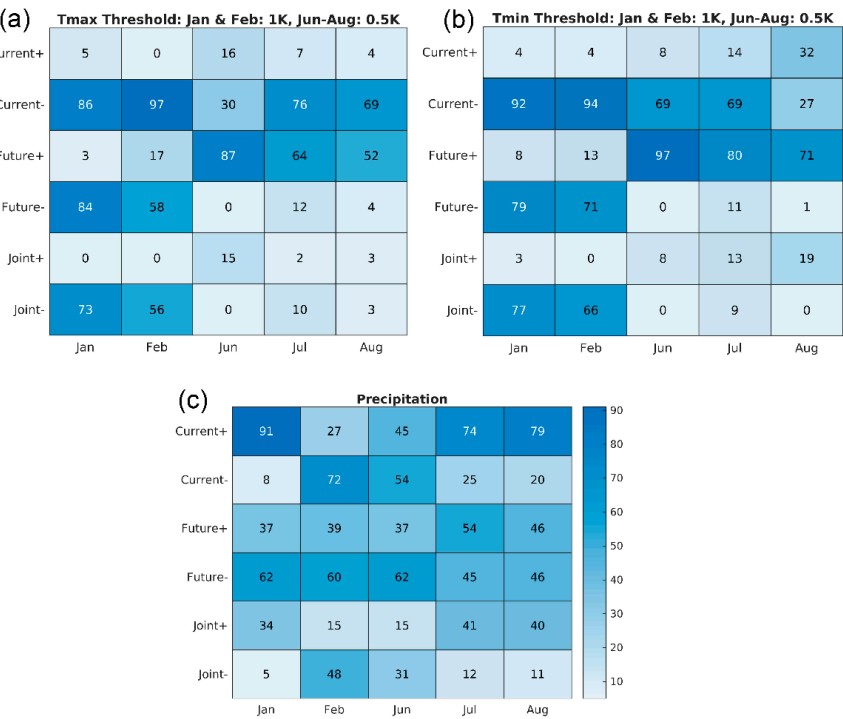

**Figure 7.** Heatmap showing the percentage of the WRF simulation domain (CONUS land only) that show positive or negative anomalies of the specified magnitude in the HadGEM-WRF historical (current) and future climate simulations, and the joint area in both the HadGEM-WRF current and future climate model output with the same sign of difference, for (**a**) Tmax, (**b**) Tmin, and (**c**) precipitation. Note that in these analyses all differences from HadGEM2-WRF were evaluated at the WRF grid resolution and that thresholds used for the Tmax and Tmin differences were different in summer (±0.5 K) and winter (±1 K).

The changes in Tmin and Tmax anomalies under the two different ENSO phases in the future climate scenario versus the historical climate are consistent with the differences in low cloud cover (Figure 8). As shown in Figure 3e, there is a projection of a reduction in the magnitude of wintertime negative Tmax differences across the southeast in the future climate simulations. This is consistent with the wintertime reduction in low cloud cover across the northern Midwest states and throughout the southeast in the future simulations. This reduction in cloud cover in strong El Niño relative to weak La Niña months leads to a relative increase in Tmax due to an increase in incoming solar radiation. The increase in magnitude of the positive Tmax and Tmin anomalies in Figures 3e and 4e are also closely associated with the late summer reduction in cloud cover. The summertime cloud cover changes indicate an increase in cloud cover in early summer throughout the southeast, while by late summer there is a reduction in cloud cover. This reduction in cloud cover is also noticeable in the northeast. Again, an increase in the amount of incoming solar radiation causes changes in the magnitude of the temperature anomalies between the historical and future years. Those same locations that exhibit negative temperature anomalies in the historical climate are projected to be positive in the future climate simulations. The inference is that the regional ENSO teleconnections from a moderate-to-strong El Niño in the future simulations leads to the reduction in low level cloud cover, thereby changing the radiation balance at the surface, which is subsequently reflected in the Tmax and Tmin anomalies observed at the surface.

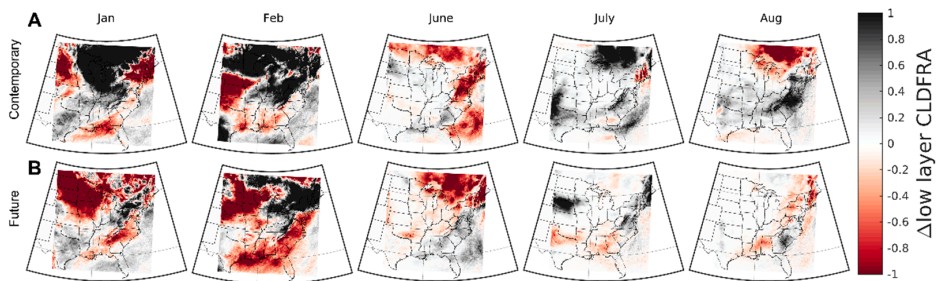

**Figure 8.** Difference in mean low level cloud cover (CLDFRA) (strong El Niño-weak La Niña) in the HadGEM2-WRF simulations for (**A**) the historical climate and (**B**) future climate scenarios, where low clouds are classified as z < 2000 m and where z is the height. Red indicates less cloud cover in the strong El Niño phase, while black indicates more cloud.

*3.4. ENSO Impacts on US Cities*

NOAA ENSO Climate Normals data for January Tmax, Tmin, and PPT are presented in [10] for various US cities. Empirical quantile–quantile plots for the 10th to 90th percentiles of HadGEM2 and HadGEM2-WRF-simulated daily Tmin and Tmax values for each day in January (i.e., coldest three days to the warmest three days) in those cities for the weak La Niña and strong El Niño conditions during the historical and future climate are shown in Figure 9. Also presented in that figure are data from National Weather Service Automated Surface Observing System (ASOS) stations (January 1985 and 2000), and the NOAA ENSO Climate Normals data for those cities also sampled for the same ENSO phases. Downscaling HadGEM2 with WRF brings the modeled values closer to the ASOS observations, although there is a notable warm bias in the HadGEM2-WRF Tmin values for Minneapolis (Figure 9). The normals from [10] also show overlap with the ASOS data, indicating that the ENSO events chosen here are not exceptional compared to the long-term NOAA ENSO Climate Normals data. A signal of higher Tmin and Tmax values under La Niña conditions are true for all cities except Boston, which is warmer under El Niño conditions. Apart from Boston, the Tmin and Tmax values on the coldest days in the La Niña January are substantially warmer than the coldest days (lowest percentiles) in the El Niño January. The same is true across the entire probability distributions for Tmax and Tmin values in the ASOS data and is also generally true for the output from the HadGEM2 and HadGEM2-WRF models. Consistent with Figures 3 and 4, the NOAA ENSO Climate Normals and the data from ASOS stations suggest that Orlando experiences higher Tmax and Tmin values when a weak La Niña is present rather than a strong El Niño phase. The HadGEM2 and HadGEM2-WRF output shows the same signal (Figure 9b). The bias in Tmin for Orlando becomes less pronounced in the upper percentiles, thus the ENSO phase mainly impacts the coldest days, but in the other cities within the model domain this is not the case. The difference in mean Tmin and Tmax (not shown) is consistent with [10], but simulations in the present study show a larger magnitude of difference for the historical climate. In addition, there is also a warm bias relative to the mean temperatures in [10]. In the future climate simulations (red diamonds in Figure 9), there is a notable increase in January Tmin and Tmax as all percentiles increase. The influence of La Niña on Tmin for Orlando shows that a warming trend continues, but La Niña impacts not only the coldest days in January but more evenly across all days. This is the case for Tmax in Orlando and Tmin in Dallas, but the other locations show that for Tmin and Tmax that ENSO-phase impacts are more notable on the warmest January days. In the case of Boston, these simulations show a reversal in the impact of ENSO, with Boston now warmer under La Niña conditions than during El Niño for both Tmax and Tmin.

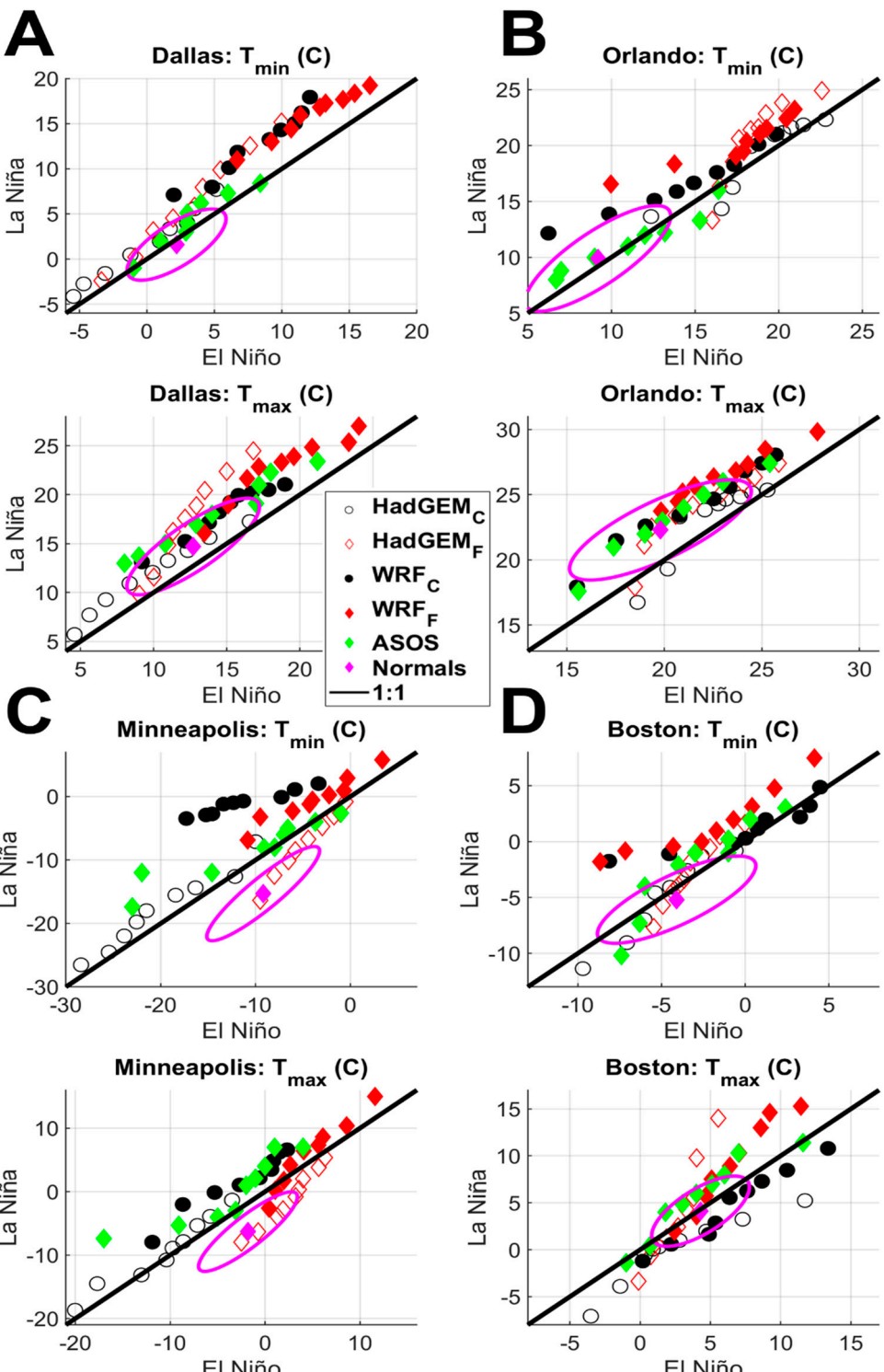

**Figure 9.** Empirical quantile–quantile plots (10th to 90th percentiles) of January Tmin and Tmax as a function of ENSO phase from HadGEM2 and HadGEM2-WRF for selected cities: (**A**) Dallas; (**B**) Orlando; (**C**) Minneapolis; (**D**) Boston. The black circles are for the historical climate and the red diamonds are for the future climate (open symbols for HadGEM2, solid for HadGEM2-WRF). ASOS observations for January 1985 (El Niño) and January 2000 (La Niña) are shown in green, the mean values for strong El Niño and weak La Niña conditions are shown by the magenta symbol, with the range shown by the ellipse [10]. The black line indicates the 1:1 slope. The points above the black line mean the La Niña days are warmer than the El Niño days.

## 4. Conclusions

There is high scientific value in seeking to improve understanding of how internal climate modes may change under externally forced climate change and how teleconnections to remote regions may be modified in the future. This research sought to address three linked questions that explored whether downscaling ESM output to drive a regional climate model modifies implied ENSO teleconnections. We performed WRF simulations at convection-permitting resolution nested with LBC from the HadGEM2 ESM. Two paired simulations were run: a strong El Niño phase and a weak La Niña phase for the historical and future years. The HadGEM2-WRF simulation output was compared to the ENSO anomalies derived from the NOAA ENSO Climate Normals dataset. We quantified ENSO teleconnections to temperature and precipitation anomalies in the historical and future climate over the entire eastern region of North America at high resolution.

The major findings of this study are as follows:

1. The regional modeling of the ENSO teleconnections with WRF using LBC from HadGEM2 show a poor degree of agreement with differences in Tmax, Tmin, and PPT values under the different ENSO phases as manifest in the observed NOAA ENSO Climate Normals data;
2. When the HadGEM2-WRF results were placed in the context of output from HadGEM2, teleconnections at the regional scale are dominated by bias in the HadGEM2 model. There was evidence to suggest that the WRF model can drive its own response to the ENSO phase. In areas of weaker signals from the LBC, the WRF generates a different regional teleconnection response;
3. A reversal in the difference in Tmax and Tmin values under different ENSO phases in the future climate relative to the historical climate was manifest in this model chain for this very high external climate forcing scenario.

However, it must be acknowledged that the reversal in the sign of temperature and precipitation differences may be symptomatic of the variability in the precise manifestations of ENSO phase [55] and of the selected simulation years. For this reason, it is not possible to definitively conclude whether the surface responses to ENSO phase may change in the future. The near-surface climate anomalies from each ENSO phase from the HadGEM2-WRF output ae, to some degree, consistent with the previously postulated eastward shift of ENSO-induced anomalies over North America [16]. For example, the region of maximum drying (Figure 5) and relative cooling (Figures 3 and 4) under the positive phase (strong El Niño) were displaced further east in winter in the future climate simulations.

Past research has sampled more future years to explore ENSO teleconnections but at lower model resolution [14,31]. Here, we applied WRF at convection-permitting resolution, which involved a significant computational burden. This meant we sampled only exemplar years of each given phase and further that we used one-way coupling between the ESM (HadGEM2) LBC and WRF and within the two WRF domains. Future studies could examine whether two-way coupling between the ESM and regional model alters the implied regional response to a given ENSO phase. Given the availability of sufficient high performance computational resources, future work to provide more concrete inferences might be derived by performing convection-permitting regional modeling using LBC from a wider suite of ESMs, a variety of external climate-forcing scenarios, and by sampling a larger number of years with differing ENSO characteristics. The sampling of a wider range of ESMs and different ENSO types is of high priority given the model-to-model variability in ENSO projections [56] and the importance of the precise spatial patterns of SSTs and induced tropical rainfall anomalies in dictating the regional responses over the North Pacific and North American climate [13].

**Author Contributions:** Conceptualization, S.C.P. and R.J.B.; methodology, S.C.P. and T.S.; model simulations, T.S.; formal analysis, S.C.P., T.S. and J.J.C.; writing—original draft preparation, T.S. and S.C.P.; writing—review and editing, T.S., S.C.P., R.J.B., and J.J.C. All authors have read and agreed to the published version of the manuscript.

**Funding:** Funding provided by the US Department of Energy (DE-SC0016438) and Cornell University's Atkinson Center for a Sustainable Future (ACSF-sp2279-2016). This research was enabled by access to computational resources supported by the NSF, ACI-1541215, and the NSF Extreme Science and Engineering Discovery Environment (XSEDE) (award TG-ATM170024), plus those of the National Energy Research Scientific Computing Center, a DOE Office of Science User Facility supported by the Office of Science of the U.S. Department of Energy under Contract No. DE-AC02-05CH11231.

**Institutional Review Board Statement:** Not applicable.

**Informed Consent Statement:** Not applicable.

**Data Availability Statement:** The NOAA-NCEP real-time global sea surface temperature analyses are available from http://www.nco.ncep.noaa.gov/pmb/products/sst/. Last accessed 15 June 2022. The NWS ASOS data are available from ftp://ftp.ncdc.noaa.gov/pub/data/asos-fivemin/. Last accessed 15 June 2022. The WRF model output generated within this project (including the name list used) are available for download at: http://portal.nersc.gov/archive/home/projects/m2645/www/public_data_hadgem_wrf Last accessed 15 June 2022.

**Acknowledgments:** We thank Anthony Arguez for providing his derived NOAA ENSO Climate Normals dataset. We thank the NSF-XSEDE and Cornell Center for Advanced Computing staff for their assistance in building and maintaining the cloud infrastructure leveraged in this research. The authors also express appreciation to the three anonymous reviewers.

**Conflicts of Interest:** The authors declare no conflict of interest. The funders had no role in the design of the study; in the collection, analyses, or interpretation of data; in the writing of the manuscript; or in the decision to publish the results.

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
