# Peer review of "Exploring ENSO-Induced Anomalies over North America in Historical and Future Climate Simulations That Use HadGEM2-ESM Output to Drive WRF"

_climate, doi:10.3390/cli10080117_

Round 1
Reviewer 1 Report
This study adopted nested WRF and ESM to investigate the ENSO influence on climate in North America in present day and future. The topic is interesting and the results are robust. I think the study should be accepted eventually in the journal. I have one concern about the modeling technique. Is the WRF regional model nested in global model with a one way coupling or two way coupling? If it is one way coupling, does the feedback from the regional model to the global model important to the results in the authors’ opinion? Based on previous studies (https://ams.confex.com/ams/91Annual/webprogram/Handout/Paper184875/seattle-2011-re4.pdf) the feedback from finer scale to the global model may be also important.
Reviewer 2 Report
Review comments for “Exploring ENSO-induced anomalies over North America in the contemporary and future climate”
This study investigates the changes in ENSO-induced anomalies in North America under global warming using a GCM and an RCM combinedly. This study could be very important to understand the changes in the extreme climate events over North America. This manuscript is well written and the figures are clear. I recommend acceptance for publication in Climate after some points clarified.
Specific comments:
1. Line 16: strong El Nino and weak La Nina -> positive El Nino and negative La Nina.
2. Lines 59-60: Some important references on the changes in ENSO’s impact were missed in the current manuscript:
Kug, J.-S., S.-I. An, Y.-G. Ham, and I.-S. Kang, 2010: Changes in El Niño and La Niña teleconnections over North Pacific–America in the global warming simulations. Theor. Appl. Climatol, 100, 275–282.
Zhou, Z.-Q., Xie, S.-P., Zheng, X.-T., Liu, Q. & Wang, H. Global warming-induced changes in El Niño teleconnections over the North Pacific and North America. J. Clim. 27, 9050–9064 (2014).
Huang, P., and S.-P. Xie, 2015: Mechanisms of change in ENSO-induced tropical Pacific rainfall variability in a warming climate. Nat. Geosci., 8, 922–926.
Hu, K., G. Huang, P. Huang, Y. Kosaka, and S.-P. Xie, 2021: Intensification of El Niño-induced atmospheric anomalies under greenhouse warming. Nat. Geosci., 14, 377–382.
Wang, Y., K. Hu, G. Huang, and W. Tao, 2021: Asymmetric impacts of El Niño and La Niña on the Pacific–North American teleconnection pattern: The role of subtropical jet stream. Environmental Research Letters, 16, 114040.
3. As reported in the literatures suggested above, a robust conclusion on the changes in ENSO-induced tropical and North Pacific anomalies is that the ENSO-induced anomalies are likely strengthen and eastward shift under global warming. This conclusion should be introduced and compared with the conclusions in this study.
4. Previous studies also suggested that the change pattern in the tropical Pacific Sea surface temperature and the changes in ENSO-induced tropical rainfall are the key process determining the changes in ENSO-related North Pacific and North America climate. Thus, I would suggest these tropical changes simulated in the selected GCM should be checked.

Reviewer 3 Report
Review of “Exploring ENSO-induced anomalies over North America in the contemporary and future climate”
Manuscript reference: climate-1817823
Authors: Tristan Shepherd, Jacob J Coburn, Rebecca J Barthelmie, Sara C
Recommendation: major revision
General Evaluation:
This work examines the performance of HadGEM2 and HadGEM2-WRF in simulating ENSO teleconnections over the Eastern US. In general, the subject is interesting and it a good supplement/extension of existing literatures However, the analysis methods are somehow weak such that significant science advances not properly presented in this manuscript. Please see comments below. I would recommend the authors to revise the manuscript before it’s suitable for publication.
Major comments:
1. it’s unclear which ENSO index is used in this study. In section 2, it says ONI is used to identify El Nino and La Nina events. However, Nino3 and Nino3.4 indices are shown in Fig.1. What’s the purpose of showing these two indices assuming ONI is the index for sampling ENSO events?
2. Figure 2, 3 and 4 show El Nino – La Nina cases. Because of the asymmetry of the ENSO teleconnections, we can gain only limited information from these difference analyses. Instead, El Nino and La Nina phases should be separated in the analysis.
3. Fig. 6 compare historical simulation vs projections. Normally comparison of this kind should be based on a large number of ENSO events. However, only two periods are included in this analysis, such that conclusions are not representative at all for describing performance of the model in simulating future vs historical ENSO.
Minor comments:
1. Model names should be added in the title, as this study is based on specific models.
2. “contemporary” is not a common term to describe past climate simulations. We use “historical” for these model simulations.
Round 2
Reviewer 3 Report
All previous comments have been satisfactorily addressed.